# Asymptotic Free energy of Variational Bayesian Deep Learning

**Hiroshi Wakimori & Tikara Hosino**
Nihon Unisys, Ltd.
1-1-1 Toyosu Koto-ku, Tokyo 135-8560 Japan

{hiroshi.wakimori,chikara.hoshino}@unisys.co.jp

## Abstract

The Bayesian deep learning is promising for its theoretical foundation. Especially, it was probed that free energy can asymptotically identify the structure of the true distribution consistently. In this paper, we derive the asymptotic expected variational free energy in the case of Gaussian trial posterior. The result shows that the variance of the posterior reflects the relative structure of the true distribution and the learning model. This result clarifies the theoretical insights of model selection and model distillation in variational approximation of Bayesian methods.

## 1 Introduction

Bayesian learning of deep neural networks has several advantages over maximum likelihood method and maximum a posteriori method (MAP). For example, the Bayesian method avoids over-fitting of models and resolves model selection problems in a theoretically established way (Watanabe, 2009). Since accurate calculation of posterior distribution using MCMC is computationally inefficient, approximation methods such as variational method are widely used. The variational method approximates the posterior distribution of parameters by the trial distribution using the variational principle on free energy.

$$\text{Free energy} \equiv -\log p(x^n) = -\log \int p(x^n|\theta)p(\theta)d\theta = -\log \int p(x^n|\theta)p(\theta)\frac{q(\theta)}{q(\theta)}d\theta$$
$$\leq E[-\log p(x^n|\theta)]_{q(\theta)} + KL(q(\theta)||p(\theta)),$$

where $x^n$ is $n$ points of sample $x$, $p(x|\theta)$ is learning model, $p(\theta)$ is prior of the parameters and $q(\theta)$ is trial posterior. In this paper, as trial posterior, we deal with the product of univariate Gaussian distribution $q(\theta) = \prod_{i=1}^{|\theta|} \mathcal{N}(\theta_i|\mu_i, \sigma_i^2)$ (Hinton & Van Camp, 1993; Graves, 2011). This approximation is implemented by Bayesian deep learning frameworks such as Edward (Tran et al., 2016). In mixture of exponential family and reduced rank regression case, the mean field type of variational free energy is clarified (Watanabe & Watanabe, 2006; Nakajima & Watanabe, 2007). Despite being widely used in practice, the theoretical properties of this approximation in deep learning have not been clarified.

The contributions of this paper are as follows. First, we derive the asymptotic expected free energy in the variational Bayesian deep learning. Second, we show that the asymptotic expected variational free energy of deep learning depends on the true distribution, unlike the regular statistical models. Third, we show that the variance of the posterior distribution reflects the relative structure of the true distribution and the learning model.

## 2 Simple Example

In this section, we present the key points of our analysis by a simple example. We consider the following learning model of which input and output is $(x, y)$.

$$p(y|x, a, b, c) = \frac{1}{(2\pi)^{\frac{1}{2}}} \exp\left(-\frac{1}{2}(y - af(bx) - cx)^2\right), p(a), p(b), p(c) \sim \mathcal{N}(0, 1)$$

where $f(x)$ is the activate function, let $f(x) = x + x^2$. To examine the behavior of the redundant part of the learning model, we assume $x$ and $y$ are irrelevant and independently generated from $p_0(x), p_0(y) \sim \mathcal{N}(0, 1)$.

For parameters, we introduce trial posterior $q(a) \sim \mathcal{N}(\mu_a, \sigma_a^2), q(b) \sim \mathcal{N}(\mu_b, \sigma_b^2), q(c) \sim \mathcal{N}(\mu_c, \sigma_c^2)$. Then, the objective function of the expected variational free energy is given by

$$objF_{vb}(n) \equiv E[E[-\log p(y^n|x^n, a, b, c)]_{q(a),q(b),q(c)}]_{p_0(x),p_0(y)} + \sum_{\theta_i \in (a,b,c)} KL(q(\theta_i)||p(\theta_i))$$

$$= nS_0 + \frac{n}{2}\left((\mu_a\mu_b + \mu_c)^2 + \mu_a^2\sigma_b^2 + \mu_b^2\sigma_a^2 + \sigma_a^2\sigma_b^2 + \sigma_c^2 + 3(\mu_a^2 + \sigma_a^2)(3\sigma_b^4 + 6\mu_b^2\sigma_b^2 + \mu_b^4)\right)$$

$$+ \frac{1}{2}\log\frac{1}{\sigma_a^2} + \frac{\sigma_a^2}{2} + \frac{\mu_a^2}{2} - \frac{1}{2} + \frac{1}{2}\log\frac{1}{\sigma_b^2} + \frac{\sigma_b^2}{2} + \frac{\mu_b^2}{2} - \frac{1}{2} + \frac{1}{2}\log\frac{1}{\sigma_c^2} + \frac{\sigma_c^2}{2} + \frac{\mu_c^2}{2} - \frac{1}{2},$$

where $S_0 = \frac{1}{2}(2\pi + 1)$ is the entropy of $p_0(y)$. We consider the minimization of this equation in the asymptotic case ($n \to \infty$). In its minimization, $\mu_a^2, \mu_b^2, \mu_c^2$ is obviously 0.

$$objF_{vb}(n) = nS_0 + \frac{n}{2}\left(\sigma_a^2\sigma_b^2 + \sigma_c^2 + 9\sigma_a^2\sigma_b^4\right) + \frac{1}{2}\left(\log\frac{1}{\sigma_a^2}\frac{1}{\sigma_b^2}\frac{1}{\sigma_c^2}\right) + \frac{\sigma_a^2}{2} + \frac{\sigma_b^2}{2} + \frac{\sigma_c^2}{2} - \frac{3}{2}$$

This equation shows that the leading-order term of $objF_{vb}(n) - nS_0$ is determined by the variance of the posterior. Thus, to determine the order of the variance, we assume $\sigma_i^2 = O(\frac{1}{n^{\alpha_i}})$.

$$objF_{vb}(n) = nS_0 + \frac{1}{2}\left(\frac{n}{n^{\alpha_a+\alpha_b}} + \frac{n}{n^{\alpha_c}} + 9\frac{n}{n^{\alpha_a+2\alpha_b}}\right)$$

$$+ \frac{1}{2}(\alpha_a + \alpha_b + \alpha_c)\log n + \frac{1}{2}\left(\frac{1}{n^{\alpha_a}} + \frac{1}{n^{\alpha_b}} + \frac{1}{n^{\alpha_c}}\right) + \text{ terms independent from } n.$$

This is the optimization problem under $0 \leq \alpha_a, \alpha_b, \alpha_c \leq 1$.

- Second term : $\alpha_a + \alpha_b \geq 1$ and $\alpha_c \geq 1$ (The variance of the posterior is upper bounded.)
- Third term : $\min \alpha_a + \alpha_b + \alpha_c$ (Maximize the entropy of the posterior.)
- Fourth term : $\min \frac{1}{n^{\alpha_a}} + \frac{1}{n^{\alpha_b}} = \max \min A$ (As possible, make the variances equally.) where $A$ is the set of $\alpha$ not determined by the above conditions.

At the first condition, we ignored $\alpha_a + 2\alpha_b \geq 1$ because it is satisfied when $\alpha_a + \alpha_b \geq 1$ is satisfied. It is noted that, in the same manner, the higher order terms than $\sigma_i^2$ can be ignored by the characteristics of the moments of the Gaussian distributions. We use this fact in the general case. It is understood that it is sufficient to solve as strong constraint conditions in order from the top. In this example, the solution is $\alpha_a = \alpha_b = \frac{1}{2}, \alpha_c = 1$ and the variational free energy is given by

$$F_{vb}(n) = nS_0 + \log n + O(1).$$

## 3 GENERAL CASE

As the general case, we assume the following conditions. First, the true distribution and the learning model are feedforward type deep neural networks. Second, the learning model includes true distribution $p(y|x)$ and the true distribution $p_0$ is parameterized by minimum number of parameters which satisfy the condition $KL(p_0(y|x)||p(y|x)) = 0$ and we define the entropy of the true distribution $S \equiv E[-\log p_0(y|x)]_{p_0(y|x)p_0(x)}$. Third, the activation function $f(x)$ has Taylor expansion and the expansion begins with $x$ term (such as $tanh$, $sigmoid$). Fourth, the prior distribution of the parameters are $\mathcal{N}(0, \sigma_0^2)$.

The general case can be solved in the same way. Let $Path$ be the set of all paths including redundant parameters on the computational graph and $\theta$ be all parameters. We consider the following terms of the objective function,

$$\frac{n}{2}\sum_{path_j \in Path}\prod_{\theta_i \in path_j}(\mu_i^2 + \sigma_i^2) + \sum_{k=1}^{|\theta|} KL(q(\theta_k)||p(\theta_k)).$$

In this case, the mean of $\theta_i$ that belongs to the true distribution is $\mu_i = \mu_i^* = $ const and the mean of redundant parameter $\theta_i$ is $\mu_i = 0$. Then, we need to solve the optimization for $\alpha_i$ ($\sigma_i^2 = O(\frac{1}{n^{\alpha_i}})$) :

- $\alpha_i = 1$ if $\theta_i$ belongs to the true distribution.
- $\forall \quad path_j \in Path \quad \sum_{\alpha_i \in path_j} g(\alpha_i) \geq 1$
  where $g(\alpha_i)$ returns 0 if $\theta_i$ belongs to the true distribution, otherwise $\alpha_i$.
- $\min \alpha_1 + \cdots + \alpha_{|\theta|}$
- $\min \frac{1}{n^{\alpha_1}} + \cdots + \frac{1}{n^{\alpha_{|\theta|}}} = \max \min A$
  where $A$ is the set of $\alpha$ not determined by the above conditions.

Then, using the solution of $\alpha_i$, the variational free energy is given by

$$F_{vb}(n) = nS + \frac{\lambda}{2} \log n + O(1), \ \lambda = \alpha_1 + \cdots + \alpha_{|\theta|}.$$

## 4 FULLY CONNECTED DEEP LEARNING

For concrete example, we consider fully connected deep neural networks. We assume the network has $L$ layers and each layer $i$ has $n_i$ nodes. Let the number of the nodes which belong to the true distribution is $t_i > 0$ and that of the other redundant nodes is $r_i \geq 0$. The output layer satisfies $t_L = n_L, r_L = 0$. In this case, by solving the above optimization, the variational free energy $F_{vb}$ (under the $n \to \infty$) is given by

$$F_{vb}(n) = nS + \frac{\lambda}{2} \log n + O(1), \ \lambda = \sum_{i=j}^{L-2}(n_i - n_{i+2})r_{i+1} + \sum_{i=1}^{L-1} n_i t_{i+1},$$

where $j = \min_{j=1}^{L-2} \sum_{i=j}^{L-2}(n_i - n_{i+2})r_{i+1}$.

Specific examples are shown in Figure 1 and 2. Figure 1 is an example in which the numbers of nodes in all layers are the same. All $\alpha$ of connections between redundant nodes are $\frac{1}{L-1} = \frac{1}{4}$. On the other hand, Figure 2 is an example in which the number of nodes in a layer is smaller than other layers. Around this layer, the symmetry breaks and $\alpha$ of connections between redundant nodes are larger than other layers. In both examples, $\alpha$ of connections between true distribution nodes are 1. Thus, in variational Bayesian learning, it is theoretically possible to distinguish redundant nodes using $\alpha$ (the variance behavior against n).

In these examples, we show that the order of the variances is not uniform but determined by the relative structure of the true distribution and the learning model (Takamatsu et al., 2006). This behavior is different from that of the regular statistical models whose variances of the posterior are uniformly $O(\frac{1}{n})$ and coefficient $\lambda$ is the number of the model parameters. Moreover, the free energy of exact Bayes differs from that of variational approximation due to the higher order terms ignored by the Gaussian approximation of the posterior distribution (Aoyagi, 2013). Finally, in these examples, the mean of the parameters $\mu_i$ is the same as MAP solution. The difference is occurred when the objective function is not expected by the true distribution. For this problem, analyzing the effect of integration by the trial posterior $E_{q(\theta)}[-\log p(y^n|x^n, \theta)]$ is the future work.

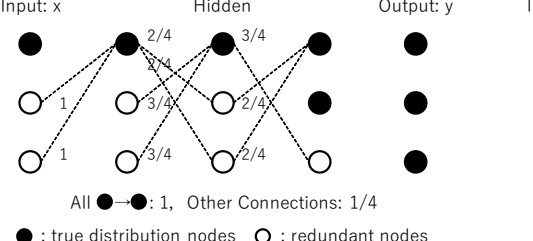

Figure 1: $\alpha$ of the example that every layer has the same number of nodes.

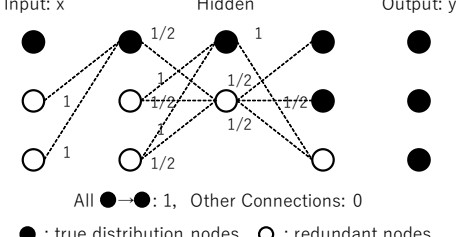

Figure 2: $\alpha$ of the example that a layer has small number of nodes.

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
