# OpenReview forum: "Asymptotic Free Energy Of Variational Bayesian Deep Learning"
_ICLR.cc/2018/Workshop — Reject_

### Official Review · AnonReviewer2 · 2018-02-23
**Interesting contribution on the asymptotic value of Gaussian variational approximation for deep**

**Rating:** 7
**Confidence:** 3

**Review:**

This is a short contribution so a lot of details are necessarily left out but I am missing more explanation on assumptions and reasons for the results and consequences of results. For example why will no higher order cumulants appear in the results.

Minor:

activate function -> activation function

---

### Official Review · AnonReviewer1 · 2018-03-06
**Motivation is weak**

**Rating:** 4
**Confidence:** 4

**Review:**

In this paper, the free energy of deep neural networks is analyzed in the asymptotic sense --- the sample size goes to infinity.

Although the goal is interesting, the motivation is unclear. The paper doesn't explain why do we need to care about the asymptotics of variational DNNs. In this paper, model selection is mentioned as an application, but no detail is provided. How can we use model selection for DNNs? For hyperparameter search? If so, how is it prominent compared to existing hyperparameter search methods?

Also, the paper doesn't contain any experimental results. I know the paper is theoretically oriented, but the experiments still help for such as sanity check of the theoretical results.

---

### Official Review · AnonReviewer3 · 2018-03-09
**The paper is hard to follow.**

**Rating:** 3
**Confidence:** 1

**Review:**

It seems that the paper is trying to study the asymptotic properties of expected variational free energy with diagonal Gaussian trial distributions and demonstrate its validity both empirically and theoretically. The clarity of this paper is poor.  Perhaps due to the limit of the space, many pieces are missing from the text. It is unclear what kinds of simplification is made to derive the conclusions and how would they generalize to guide the practice. For example, what is the "relative structure"? Why is it theoretically possible to distinguish redundant nodes using alpha? To simplify the derivation, the authors made a list of assumptions, it is unclear how are they satisfied in practice. With important technical and experimental details missing, I think this paper is a clear rejection.

---

### Decision · Program_Chairs · 2018-03-20
**ICLR 2018 Workshop Acceptance Decision**

**Decision:**

Reject

**Comment:**

Based on the reviews, this paper has not been accepted for presentation at the ICLR workshop. However, the conversation and updates can continue to appear here on OpenReview.